

# Fast grassland recovery from viable propagules after reintroducing traditional mowing management on a steep slope

Susumu Yamada[1], Wakana Yoshida[1], Minori Iida[1], Yoshiko Kitagawa[2] and Jonathan Mitchley[3]

[1] Tokyo University of Agriculture, Atsugi, Kanagawa, Japan
[2] Tama Kyuryo Sha, Yokohama, Kanagawa, Japan
[3] School of Biological Sciences, University of Reading, Reading, United Kingdom

## ABSTRACT

Semi-natural grasslands on steep slopes often show high plant species diversity. These grasslands were traditionally maintained through mowing and/or grazing. The traditional management practices help to maintain species diversity, whereas land abandonment reduces diversity by increasing competition from dominant species and reducing seedling recruitment. The reintroduction of management can reverse species diversity declines, but suitable grassland restoration programs are scarce in Japan. To study the effect of short-term abandonment on seedling ecology, we monitored the vegetation of a Susogari grassland that had been abandoned for 3 years; the grassland occupies a steep slope (ca. 50°) on a hillside above paddy fields, and was traditionally mown. We monitored the vegetation before abandonment, in the 3rd year of abandonment, and in the 1st and 2nd years after restoration of mowing management. Emergence and survival of seedlings was monitored for 18 months after reintroduction of management. We monitored 1,183 seedlings of grassland species and non-target annuals in ten 1-m$^2$ plots. After mowing was reintroduced, most grassland species reappeared or increased in the first and second years. Few seedlings of perennial plants and no seedlings of annuals flowered. An exotic species, *Solidago altissima*, had a lower survival rate (10%) than grassland species (>30%), and all but two grassland species survived over the 18-month period. Although vegetation composition was not fully recovered, our findings suggest that a steep slope acts as a strong filter that inhibits the establishment of non-target species while enhancing persistence of target grassland species.

Corresponding author
Susumu Yamada,
sy206447@nodai.ac.jp

## INTRODUCTION

Semi-natural grasslands exist as a result of human activity (mowing and/or livestock grazing) in artificially deforested areas (*Raduła et al., 2020*). Semi-natural grasslands provide key areas for maintaining biodiversity in agricultural landscapes (*Duelli & Obrist, 2003*; *Johansen, Henriksen & Wehn, 2022*). In these fragmented landscapes, remnant habitats such as field margins, ditches, stone walls, midfield islets, hedgerows, and ponds

also harbor many grassland species (*Marshall & Moonen, 2002*; *Lindborg et al., 2014*). Fragmented habitats usually have high rates of input of propagules of habitat non-specialist species from the surrounding area, which increases the probability of invasion by non-specialists, resulting in different floristic composition from typical semi-natural grasslands (*Fischer & Stöcklin, 1997*; *Boutin & Jobin, 1998*; *Lindborg et al., 2014*). The decline of semi-natural grassland areas and the species dependent on them increases the importance of surrogate habitats that can substitute for semi-natural grasslands (*Duelli & Obrist, 2003*). This is particularly the case where natural grasslands are rare.

Remnant habitats such as midfield islets and linear grasslands on steep slopes are associated with land that is sub-optimal for agriculture owing to poor abiotic conditions (*i.e.,* steep ground or shallow soils) (*Lindborg et al., 2014*; *Johansen, Henriksen & Wehn, 2022*). They were mown to prevent shading of the nearby fields or were grazed by low intensity livestock. These remnant habitats can support high species richness. For instance, slope steepness often enhances the diversity of grassland species, which highlights the disproportionate conservation value of these sites and their importance as biodiversity hotspots (*Luoto, 2000*; *Pykälä et al., 2005*; *Bennie et al., 2006*). Steep grasslands are also more resistant to invasion by competitive species than flatter sites owing to nutrient limitation, water runoff, and soil disturbances (*Pykälä et al., 2005*; *Bennie et al., 2006*). These unusual topsoil conditions result in more stressed habitat conditions than in the surrounding areas (*Loydi et al., 2013*). Lower coverage of competitive species promotes seedling survival by providing a greater number of favorable microsites (*Fowler, 1988*; *Isselstein, Tallowin & Smith, 2002*).

Abandonment of traditional agricultural practices has been a major threat to maintaining such remnant habitats in most marginal regions of Europe (*Poschlod, Bakker & Kahmen, 2005*; *Stoate et al. 2008*) and in East Asia (*Nakamura & Short, 2001*; *Chen et al., 2021*). Although successional patterns vary according to climate and landscape conditions (*Prévosto et al. 2011*; *Pérez-Hernández & Gavilán, 2021*), most studies noted a decrease of specialist species in grasslands after abandonment (*e.g.*, *Rosenthal, 2003*; *Török et al., 2008*). The specialist species contribute to species richness at the regional level, so this decline of specialists has motivated attempts to improve the diversity of typical grassland species in abandoned semi-natural grasslands (*Ruprecht, 2006*; *Waldén et al., 2017*).

The reintroduction of former management regimes is a widely used restoration measure (*Ruprecht, 2006*; *Waldén et al., 2017*). Both the availability and the persistence of target species are key aspects of grassland restoration success (*Hutchings & Booth, 1996a*; *Hutchings & Booth, 1996b*). First, the successful reconstitution of the vegetation community depends on seed availability in the soil seed bank, seed rain (*Zobel, 1997*; *Stampfli & Zeiter, 2020*) and vegetative reproduction (*Touzard et al., 2002*; *Matus, Tóthmérész & Papp, 2003*). If most floristic components have persisted in the vegetation or soil seed banks, then spontaneous succession of the former vegetation is likely when the natural disturbance regime is reinitiated (*Prach, 2003*). However, if viable propagules no longer exist, due to its being too long since abandonment, spontaneous restoration would be difficult (*Rosef, 2008*; *Raduła et al., 2020*).

The seedling stage is the most vulnerable part of the plant life cycle (*Stebbins, 1971*; *Fenner & Thompson, 2005*; *Larson et al., 2015*). Seedlings require specific conditions for successful establishment and subsequent growth (*Grubb, 1977*), and they are more sensitive to environmental conditions (*e.g.,* desiccation and frost) and competition than mature plants (*Turnbull, Crawley & Rees, 2000*; *Leger, Atwater & James, 2019*). As a result, seedling mortality is exceptionally high, sometimes even reaching 100% (*Silvertown & Dickie, 1980*; *Heinken-Šmídová & Münzbergová, 2012*; *Larson et al., 2020*). In addition, the germination success of most species is reduced by competition with established plants (*Fenner, 1978*; *Yamada, Nemoto & Okuro, 2021*). Thus, it is necessary to monitor individual seedlings for at least a year to determine their fate and whether plant communities are successfully re-established (*Fenner, 1978*; *Stampfli & Zeiter, 1999*; *Yamada, Nemoto & Okuro, 2021*).

Semi-natural grasslands are important at the national level in Japan (*Ministry of the Environment Japan, 2024*). Studies have shown that small grasslands on steep slopes (ca. 50°) are key elements for conserving regional floristic diversity in agricultural landscapes (*Kitazawa & Ohsawa, 2002*; *Okubo et al., 2005*). Herbaceous vegetation on steep slopes is widespread in Japan (*Uematsu & Ushimaru, 2013*) and Southeastern Asia (*Kumalasari & Bergmeier, 2014*). Despite the fact that many such grasslands are threatened by land abandonment, relatively few restoration projects have been undertaken. *Yamada (2008)* reported that if a grassland had been unmanaged for more than 10 years, most grassland species did not become re-established, even if former management (mowing) was reintroduced. If the duration of abandonment is shorter, however, it is more likely that propagules (seed bank and sprouts) will be viable in the soil. If so, management reintroduction will be less costly and more likely to lead to successful restoration of native communities (*Metsoja et al., 2012*). However, seeds and soils tend to run off quickly on steep slopes, making it more difficult to restore the former grassland community. Indeed, *Koyanagi et al. (2008)* demonstrated that a steep grassland had only a small number of seeds of grassland species. Another possibility is that restoration would increase after land abandonment due to the buildup of vegetation (*Zuazo & Pleguezuelo, 2008*). Thus, it remains unclear whether grasslands on steep slopes can be restored successfully even after a short period (<10 years) of abandonment.

This case study was designed to evaluate the restoration of traditionally mown grassland vegetation on a steep slope restored after 3 years of abandonment. The restoration method was reintroduction of mechanical mowing. We monitored vegetation in three stages: pre-abandonment, during abandonment, and during restoration. We also monitored the fate of seedlings for 18 months after the reintroduction of mowing. We hypothesized that reintroduction of mowing shortly after grassland abandonment would stimulate spontaneous recruitment of native species from the propagules (seed bank and sprouts). We expect that the number of grassland specialists and the numbers of other functional groups would be recovered to the pre-abandonment level. The criteria of restoration success should be clearly established to evaluate restoration projects. *Ruiz-Jaén & Aide (2005)* encouraged restoration projects to include at least two variables within each of the three ecosystem attributes; diversity, vegetation structure, and ecological processes. We aimed to evaluate restoration success through diversity and vegetation structure.

## MATERIALS & METHODS

Portions of this text were previously published as part of a preprint (https://www.researchsquare.com/article/rs-1857617/v1).

### Study area

The study area is located in the Tama Hills, 30 km west of Tokyo (35°35′N, 139°25′E; Fig. 1). The mean annual precipitation at the nearby Hachioji meteorological station (13 km northwest of the study area) is 1,540 mm. The mean annual temperature is 14.8 °C, with a mean minimum of 3.4 °C (January) and mean maximum of 26.4 °C (August) (*Japan Meteorological Agency, 2024*). Monthly precipitation and temperature in 2008, 2018, 2019 and 2020 are illustrated in Table S1; precipitation in January and February 2019 and August 2020 was less than the monthly average.

In the Tama Hills, hill ridges are about 30 m higher than the adjacent valley bottom. The valley bottoms are used as paddy fields, and ridges and hillsides are used for coppiced woodlands (Fig. 1B), a land use pattern that is a typical feature of Japanese agricultural landscapes. The valley bottoms have good water availability for rice culture, but the shade from adjacent ridges and their standing trees in the narrow valleys is disadvantageous. To improve light availability, standing vegetation has been periodically mown on the lower slopes (so-called Susogari grassland) (Fig. 1D). In the study area, the lowermost slopes have traditionally been mown two or three times a year (*Okubo et al., 2005*; *Yamada et al., 2005*). Grassland habitats are rare in the study area, so grassland specialists are important for the conservation viewpoint, even if they are not listed as threatened species (*Yamada, 2011*).

The study grassland is on a southeast-facing slope (S45°E) at a mean angle of 52.6° ± 7.2° (SD), and the slope was mown in a band about 7 m wide. The woodland upslope of the grassland is dominated by the deciduous broad-leaved species *Quercus serrata*, *Quercus acutissima*, and *Castanea crenata*, formerly coppiced but left unmanaged here for about 50 years. Rice culture in the valley bottom adjacent to the study plots was carried out until 2015. and abandoned since then, when mowing also stopped. The field monitoring was permitted by the Tama Environment Office of the Bureau of Environment, Tokyo Metropolitan Government.

### Reintroduction of management

The study area lies inside the 33-ha Zushi-Onoji Historical and Environmental Conservation Area, which was proclaimed a Greenery Designated Conservation Area by the Tokyo Metropolis in 1978. The government entrusted the management to the residents, who established the privately run Machida Historic Environment Conservation Management Organization. Since 1997, the organization has managed the abandoned land as was previously done, aiming to restore the former agricultural landscapes in the area. The organization clear-cut 0.4 ha of the woodland in November 2017, when the slope below was covered with tall herbaceous vegetation. In October 2018, the slope was mown after having a gap of 3 years. Since then, the site was periodically mown (*i.e.,* on 7 June

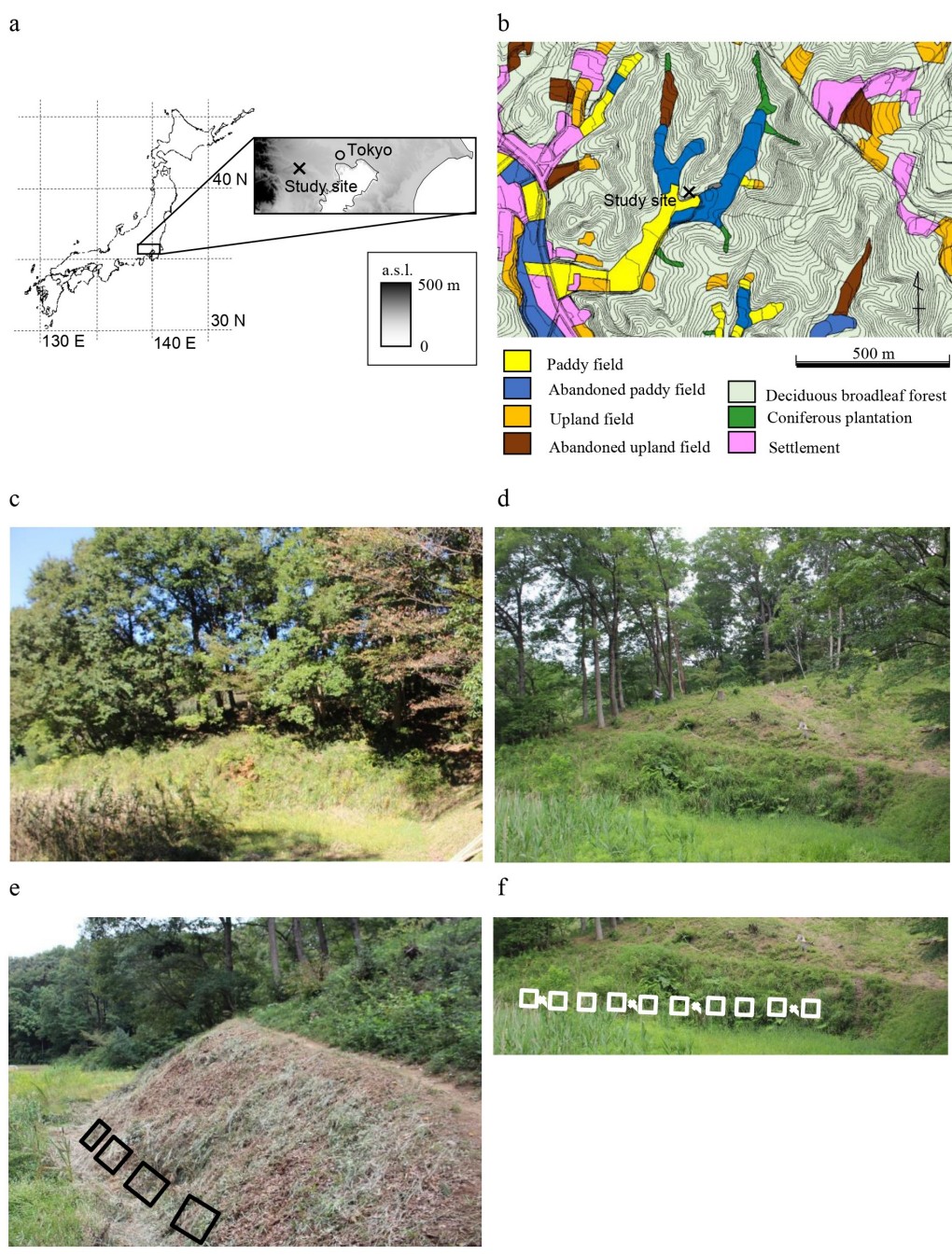

**Figure 1  Description of the study area.** (A) Location of the study site; (B) land use map of the study area; (C) photo taken before the clear-cut of the upslope woodland (October 2017); (D) photo taken after the clear-cut of the upslope woodland and before the reintroduction of mowing (May 2018); (E) photo taken after the reintroduction of mowing (October 2018) and layout of □ survey plots; (F) layout of □ the 10 plots and × soil sampling points (added to the photo taken in May 2018). Maps in (A) and (B) were published in *Yamada, Kitagawa & Okubo (2013)*.

2019, 7 August 2019, 2 June 2020, and 5 August 2020) by the organization or some authors (Yamada S, Yoshida W and Iida M).

## Monitoring

We monitored vegetation in 2008, a period under continuous management; in 2018, 3 years after the management was terminated; and in 2019 and 2020, the first and second years of management reintroduction. This grassland is organized into relatively small, discrete linear elements because of the abrupt slope angle; hence, we established ten 1-m × 1-m plots (quadrats) along a transect at intervals of 1 m (*i.e.,* a total of 19-m length, with an orientation of S 45°E). In each plot, we visually estimated the percentage cover of each species on 28 June 2008, 26 June 2018, 28 June 2019, and 2 June 2020. Small individuals with cotyledons could not be identified in 2008, because it is difficult to identify most of the early stage of seedlings without monitoring later growth. In 2019 and 2020, several seedlings recorded in the seedling emergence survey (see below) were not recorded in the vegetation survey. In 2019, many seedlings and sprouts (vegetative regrowth) began to grow, both of which are assumed to affect subsequent vegetation development, so in June 2019, each perennial and woody species was recorded along with its growth stage (seedling or sprout). In addition, vegetation cover was visually estimated on 10 May 2019, 5 August 2019, 4 May 2020, and 7 August 2020 to reveal the relationship between vegetation cover and seedling survival.

Seedling emergence and survival was monitored once a month from May to October in 2019 and from April to October in 2020. To distinguish the contribution of germination and subsequent survival to the overall rate of establishment, allowing seedlings to become a permanent part of the vegetation, we marked every germinated individual with a numbered plastic tag. To assess seedling recruitment, we identified recruited seedlings as those that still had their cotyledons and were not recorded in a previous census. Individual seedlings were then monitored in subsequent censuses. We were unable to find about a third of the tags owing to severe natural soil disturbance at the site. Because we were unsure of whether the plants were still alive nearby without tags or had died, we defined them as disappeared and eliminated them from the calculation of survival rates. Flowering of individual plants, if any, was recorded.

Surface soil was sampled to a depth of five cm with a 4-cm-diameter core sampler on 25 December 2020. Sampling was conducted at four points where the vegetation survey was not done (see Fig. 1); the center and four points equidistant from the center and each edge of the 1-m × 1-m unsurveyed points were sampled.

In early October 2020, light conditions were measured in the center of quadrats; a camera with a 180° hemispherical lens (Nikkor 8-mm f/2.8 fisheye lens) was positioned facing upward at 50 cm above the ground. Relative solar radiation was calculated in Gap Light Analyzer v.2.0 software (*Frazer, Canham & Lertzman, 1999*).

## Analysis

Following *Chibaken-shiryou-kenkyu zaidan (2003)*, all species were classified according to their life histories: woody species; annual or biennial species (hereafter "annuals"); and

perennial herb species, distinguished between small-statured plants (<30 cm) and larger perennials. Plant height classification was applied only to perennials, according to the reference above. Characteristic species representing grassland vegetation in Miscanthetea sinensis communities were defined as ''typical grassland species'' (*Miyawaki, 1994*). Typical grassland species were further divided into dominant matrix forming grasses, middle-statured subordinate perennial forbs and small-statured subordinate perennial forbs, according to their behavior in the vegetation. Nomenclature is based on BG Plants (http://ylist.info).

About half the seedlings in the genus *Cirsium* were identified as either *C. oligophyllum* or *C. japonicum*, whereas other *Cirsium* seedlings died before identification. According to the flora list in the conservation area, unidentified *Cirsium* seedlings were likely to be either *C. oligophyllum* or *C. japonicum*; thus, the three categories were combined, and survival rates are given for *Cirsium* spp. The two *Cirsium* species represent semi-natural grasslands (characteristic species in Miscanthetea sinensis communities; (*Miyawaki, 1994*). We consider the establishment of *Cirsium* spp. to be informative for understanding the fate of typical grassland species. *Carex* spp. were assumed to be one of four taxa: *C. lenta* var. *lenta*, *C. lanceolata*, *C. leucochlora* var. *candolleana*, and *C. tristachya*. No seedlings could be identified to species level, so all seedlings were combined and reported at the genus level as *Carex* species.

Established vegetation communities in 2008, 2018, 2019, and 2020 were compared by non-metric multidimensional scaling (NMDS) analyses to identify any differences in species composition between years. The percentage cover of each species (species level data) was square-root transformed for the NMDS, which was conducted in PC-ORD for Windows *v.* 6 software (*McCune & Mefford, 1999*).

### Evaluation of restoration success

Among variables which *Ruiz-Jaén & Aide (2005)* encourages as the criteria of restoration success, we used species richness as an indicator of diversity, and score of NMDS axes which included vegetation cover as an indicator of vegetation structure. We hypothesized that if vegetation recovered in 2019 and 2020, the first and second years of management reintroduction, differences in the numbers of species in each functional category, and NMDS scores in 2019 and 2020 are indistinct from those before abandonment.

## RESULTS

### Abiotic properties

The relative solar radiation was 66.6% ± 8.8% indicating bright conditions. The nitrate-N content in the surface soil was 0.5 mg/100 g, available P content was <1.0 mg/100 g, and soil pH was 6.1, indicating nutrient-poor conditions.

### Established vegetation

Throughout the monitoring no threatened species were recorded.

In 2008, *Houttuynia cordata* was dominant (Table 1). The total number of species was highest at scales of 1 m$^2$ and 10 m$^2$ (Table 2, Fig. 2). The number of typical grassland
**Table 1  The major dominant species in each year.**

| Year | Species | Frequency |
|------|---------|-----------|
| 2008 | *Houttuynia cordata* | 0.4 |
| 2018 | *Miscanthus sinensis* | 0.7 |
|      | *Osmunda japonica* | 0.2 |
| 2019 | – | – |
| 2020 | *Miscanthus sinensis* | 0.3 |
|      | *Pleioblastus chino* | 0.2 |

**Notes.**
Dominant species were defined as those observed at least twice in 10 plots with cover of ≥20%. Frequency is the number of plots in which each species was dominant out of 10 plots.

**Table 2  Changes in species richness at a sampling scale of 10 m2 according to species attributes.**

|  | 2008 | 2018 | 2019 | 2020 |
|--|------|------|------|------|
| Total | 69 | 36 | 47 | 59 |
| Life history |  |  |  |  |
|    Annuals | 6 | 0 | 2 | 5 |
|    Perennials | 54 | 32 | 38 | 49 |
|    Woody species | 9 | 4 | 7 | 5 |
| Typical grassland species | 13 | 11 | 12 | 14 |
| Small-statured perennials | 9 | 1 | 1 | 5 |
| Ferns | 10 | 5 | 5 | 6 |
| Exotic species | 2 | 0 | 1 | 2 |

species was highest at a scale of 1 m$^2$, but it remained fairly constant over time at a scale of 10 m$^2$. Only two exotic species were observed at a scale of 10 m$^2$.

In 2018, *Miscanthus sinensis* was dominant at high frequency, followed by *Pleioblastus chino* (Table 1). The total number of species was lowest (Table 2, Fig. 2). The number of typical grassland species was lowest at a scale of 1 m$^2$.

No single species was dominant in 2019 (Table 1). The total number of species was recovered, but was still smaller than that in 2008 at scales of 1 m$^2$ and 10 m$^2$ (Table 2, Fig. 2). The number of typical grassland species recovered at a scale of 1 m$^2$.

*M. sinensis* was again dominant in 2020. The total number of species was similar in 2020 and 2008 at a scale of 1 m$^2$ (Table 2, Fig. 2). At a sampling scale of 10 m$^2$, the number in 2020 was still smaller than that in 2008. Frequency of small statured perennials of non-typical grassland species was still smaller in 2020 than that in 2008 (Fig. 3, Table S1). The number of exotic species remained small.

The results of the NMDS illustrate the differences in species composition among years (Fig. 4). Plots sampled in 2008 cluster at the left of Axis 1, whereas plots sampled in 2018 are clustered at the right. Plots sampled in 2020 generally located closer to those in 2008 than in 2019, but not fully overlapping those in 2008.

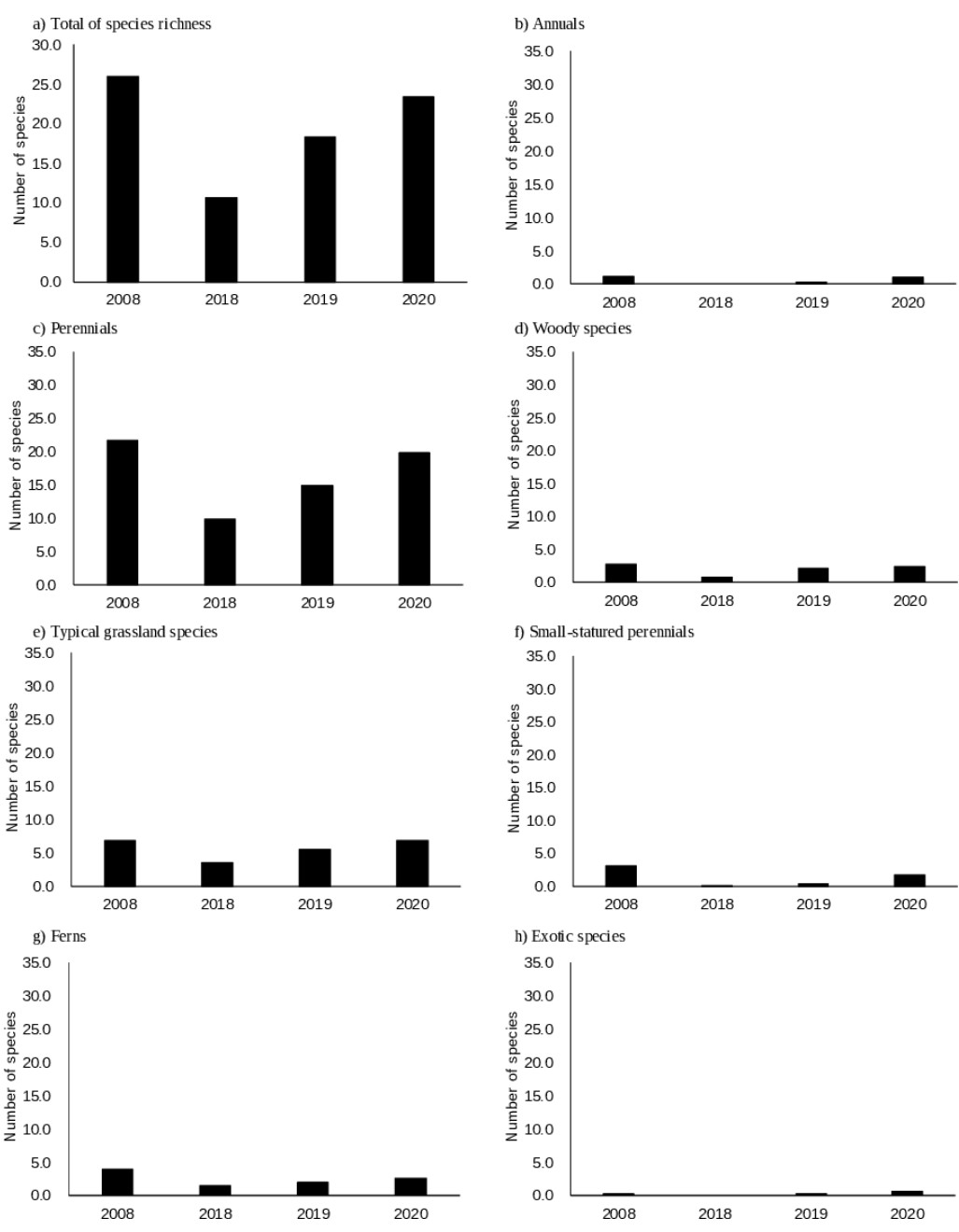

**Figure 2** (A–H) Changes in species richness at a sampling scale of 1 m² according to functional groups.

## Seedling establishment

We observed 1,183 seedlings in total over the 2-year survey. Of these, 63 were not identified and 58 were identified as either *Cirsium* spp. or *Carex* spp. Fifty-four species were recorded as seedlings, among which 17 were not recorded in the established vegetation in 2019 or 2020. All but 2 typical grassland species (*Imperata cylindrica* and *P. chino*) observed in established vegetation were recorded as seedlings (Table S1). In 2019, 615 seedlings in 46

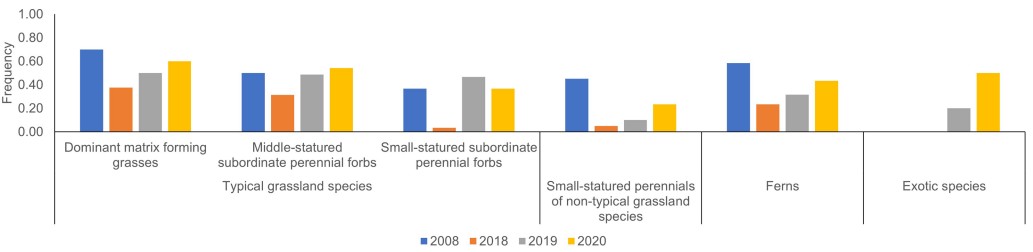

**Figure 3  Annual changes in average frequency of major species according to functional groups.** Individual species and detailed description for the criteria of major species are shown in Table S2.

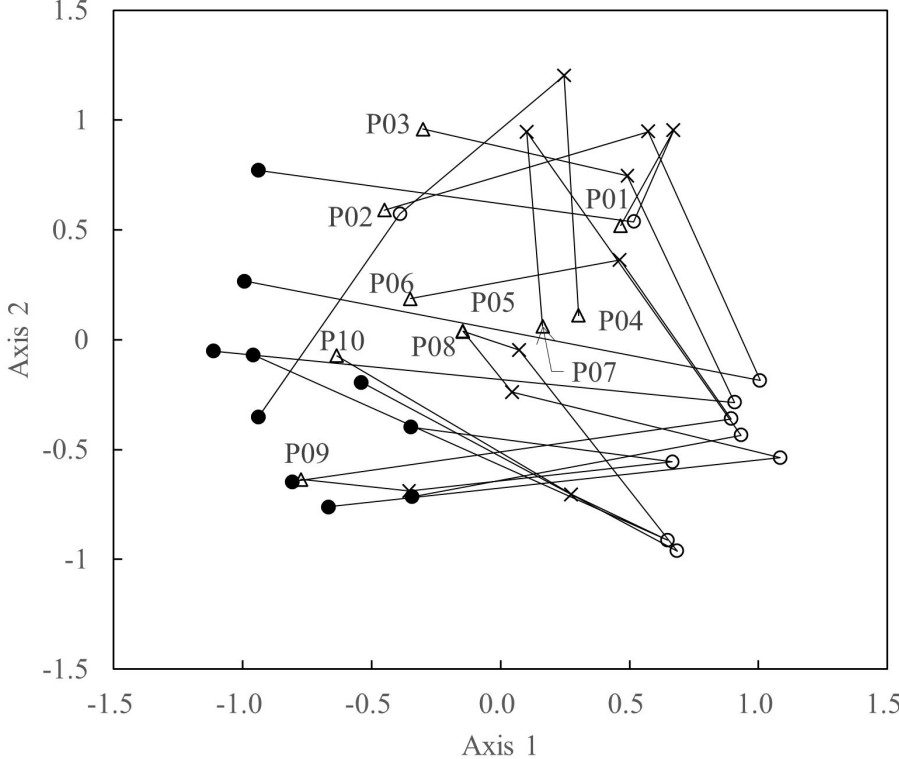

**Figure 4  Scatterplots of the first two NMDS axes in each survey period.** ● = 2008; ○ = 2018; × = 2019; △, 2020. "P01–10" = quadrat numbers.

species were recorded; the most abundant was *Thalictrum minus* var. *hypoleucum* (166), followed by *Veronica persica* (53) and *Youngia japonica* (36). In 2019, more perennial seedlings were recorded than annual seedlings. In 2020, 568 seedlings were observed; *Y. japonica* was the most abundant (271), followed by *V. persica* (67) and *T. minus* var. *hypoleucum* (46), and annual seedlings were more prevalent than perennial species.

The 1-year survival rate was >50% in *T. minus* var. *hypoleucum*, *Cirsium* ssp., and *Salvia japonica* and <20% in *Lysimachia clethroides* (Table 3). The two-season survival rate was markedly lower than the 1-year survival rate in *Solidago altissima*. We found a relatively

**Table 3  Survival rates of each species.** (a) Survival rates of nine major species and two genera. One-year survival rate, survival of seedlings for 1 year after initial emergence; two-season survival rate, survival of seedlings at the end of the second year irrespective of the month of germination in the first year; oneseason survival rate, survival of seedlings at the end of year of germination irrespective of month of germination. (b) Number of species not listed in a and found as seedlings.

**(a) Survival rates of nine major species and two genera**

| | 1-yr survival | Two-season survival rate | One season survival rate | |
|---|---|---|---|---|
| **Species** | | (October 2020) | (October 2019) | (October 2020) |
| Typical grassland species | | | | |
| *Thalictrum minus* var. *hypoleucum* | 0.571 (84/147) | 0.375 (54/144) | 0.634 (85/134) | 0.367 (11/30) |
| *Cirsium* spp. | 0.714 (20/28) | 0.370 (10/27) | 0.778 (21/27) | 0.889 (16/18) |
| *Euphorbia lasiocaula* | 0.571 (8/14) | 0.308 (4/13) | 0.571 (8/14) | 0.375 (3/8) |
| *Lysimachia clethroides* | 0.167 (1/6) | 0.000 (0/6) | 0.333 (2/6) | 1.000 (1/1) |
| *Potentilla freyniana* | 0.667 (4/6) | 0.500 (2/4) | 0.800 (4/5) | 0.857 (6/7) |
| *Miscanthus sinensis* | 0.667 (6/9) | 0.111 (1/9) | 0.667 (6/9) | – |
| *Polygala japonica* | 1.000 (9/9) | 0.667 (4/6) | 1.000 (9/9) | – |
| Other species | | | | |
| *Solidago altissima* | 0.531 (17/32) | 0.172 (5/29) | 0.806 (25/31) | 0.429 (9/21) |
| *Lysimachia japonica* | 0.600 (18/30) | 0.522 (12/23) | 0.782 (18/23) | 0.650 (13/20) |
| *Salvia japonica* | 0.583 (7/12) | 0.500 (6/12) | 0.700 (7/7) | 0.875 (7/8) |
| *Carex* spp. | 0.400 (4/10) | 0.400 (4/10) | 0.727 (8/11) | 1.000 (1/1) |
| *Ixeridium dentatum* subsp. *dentatum* | 0.444 (4/9) | 0.222 (2/9) | 1.000 (3/3) | 0.000 (0/2) |

**(b) Number of species not listed in *a* and found as seedlings**

| Species | Number of species | | Number of individuals | |
|---|---|---|---|---|
| | **Survival** | **Death** | **Survival** | **Death** |
| Typical grassland species | 1 | 2 | 1 | 3 |
| Others | 6 | 7 | 14 | 23 |

small number of seedlings of 3 typical grassland species (*Gentiana scabra* var. *buergeri*, *Sanguisorba officinalis* and *Solidago virgaurea* subsp. *asiatica*; Table 3B and Table S1). Two seedlings of *G. scabra* var. *buergeri* and one seedling of *S. officinalis* were recorded, and they died within a year. The timings of germination and mortality are shown in Fig. 4. The germination rate of perennials was particularly high up to August 2019, whereas that of annuals was high in September 2019 and from July 2020 onward. The mortality rate was large in August 2019 and in April, September, and October 2020.

Eight individuals from six species flowered during the study (Table 4).

## DISCUSSION

### Restoration outcome

Three years of abandonment led to a distinct decline in the total number of species in the established vegetation (Table 2, Fig. 2). Numbers of annual species, small-statured species, and ferns also declined markedly. Declines in small species and annuals are common in the process of secondary succession in semi-natural grasslands (*Rosenthal, 2003*; *Pykälä, 2004*;

**Table 4  Number of seedlings with inflorescences and total number of seedlings.**

| Species | Number of seedlings with inflorescences | Total number of seedlings |
|---|---|---|
| Annuals | | |
| *Pseudognaphalium affine* | 3 | 3 |
| *Youngia japonica* | 1 | 307 |
| *Picris hieracioides* subsp. *japonica* | 1 | 12 |
| Perennials | | |
| *Polygala japonica* | 1 | 12 |
| *Ajuga decumbens* | 1 | 8 |
| *Carpesium glossophyllum* | 1 | 1 |

*Lauterbach et al., 2013*). The number of species of typical grassland plants was markedly smaller in 2018 than in 2008 at a sampling scale of 1 m$^2$, but was similar between years at a scale of 10 m$^2$. For instance, *Cirsium oligophyllum* was present with higher frequencies in 2008 than in 2018 (Table S2). Thus, 3 years of abandonment led to a decline in the density of typical grassland species but did not lead to loss of most species at the community level. It usually takes more than a decade for most grassland species to disappear in the process of secondary succession (*Kahmen, Poschlod & Schreiber, 2002*; *Rosenthal, 2003*; *Török et al., 2008*).

More importantly, reintroduction of mowing contributed to rapid increases in the number of typical grassland species at the 1-m$^2$ scale (Fig. 2). Accordingly, the frequencies of typical grassland species were distinctly larger in 2019 or 2020 than in 2018 irrespective to their functional groups (Fig. 3). One reason for the fast recovery is obviously the persistence of soil seed banks. Several species observed in the present study (c.f., *Lysimachia japonica* and *Potentilla freyniana*) were reported to have persistent seed banks (*Koyanagi et al., 2011*), so 3-year-abandonment would be too short for the loss of viable seeds of most typical species. Seedling density was lower here than in grasslands in Europe (*Stampfli & Zeiter, 1999*; *Kladivová & Münzbergová, 2016*). However, seedling density was larger in the present study than in another Japanese grassland—a regularly mown grassland in Chiba Prefecture—which suggested that the density of seeds is small in managed grassland on steep slopes (*Koyanagi et al., 2008*). One possible reason for this is the persistence of several tall grassland species in established vegetation in the abandoned phase (*e.g.*, *T. minus* var. *hypoleucum*). *Yamada & Minami (2015)* demonstrated that mowing twice a year significantly decreased seed production of *T. minus* var. *hypoleucum* relative to mowing once a year, and tall grassland species benefit from low-intensity mowing (*e.g.*, biannual mowing, *Bricca et al., 2020*). Thus, as a restoration management practice, abandonment for a few years would be more advantageous for them to produce seeds than mowing twice a year, and would in turn increase the density of viable seeds. Interestingly, seedlings of the low-statured *Polygala japonica* and *Potentilla freyniana* had distinctly higher frequencies in 2020 than in 2018 (Table S1). The present study did not evaluate the seed bank nor dispersal from nearby sources. Nevertheless, since their frequencies were limited during the abandoned phase (Table S2), little seed rain would be expected during that period,

indicating that seeds persisted in the soil without being lost through soil runoff for 3 years at the site.

Although the total numbers of species increased from 2019 to 2020 at a sampling scale of 10 m² (Table 2), the number in 2020 was still smaller than that in 2008 (Table 2). Numbers of small species and ferns in 2020 were also smaller than those in 2008 (Figs. 2 and 3). Most of small species and ferns unique in 2008 (c.f., *Ajuga decumbens*, *Hydrocotyle ramiflora*, *Thelypteris japonica* and *Dryopteris erythrosora*) were observed at low frequencies (*i.e.,* 0.1 or 0.2). An originally sparse population would lead to low probabilities of establishment of seedlings at the restored site. Abandonment of extensive land use practices causes a decline of short species more than of tall species (*Rosenthal, 2003*; *Pykälä, 2004*). Although several small grassland species did not decrease (*i.e., P. japonica* and *P. freyniana*), some other small species (*e.g., Galium gracilens* and *Viola grypoceras* var. *grypoceras*) would be vulnerable to abandonment. The number of ferns did not increase from 2018 to 2020 at a sampling scale of 10 m² (Table 2). As the restored site has bright conditions, it may never be wet enough for several fern species, as suggested by *Ghorbani et al. (2003)*. In addition, surveying ferns can be difficult, because it takes longer for tiny thalli and sori of several species to grow enough to identify *via* the anterior lobe phase. Thus, it remains unclear whether quick recovery is likely or not for ferns.

### Seed recruitment and seedling survival

As expected, the 1-year seedling survival rate was greater than that in more level grasslands (*Silvertown & Dickie, 1980*; *Ryser, 1993*; *Stampfli & Zeiter, 1999*). They were particularly high (>50%) in *T. minus* var. *hypoleucum* and *Cirsium* spp. (Table 3). Previous studies showed that the unusual topsoil conditions of steep slopes (nutrient limitation, water runoff, and soil disturbances) would limit the performance of dominant species (*Pykälä et al., 2005*; *Bennie et al., 2006*) and promote seedling survival (*Fowler, 1988*; *Isselstein, Tallowin & Smith, 2002*).

The survival rates of several species were very low: the 1-year survival rate of *Lysimachia clethroides* was <20%, and the two-season survival rates of *S. altissima* and *M. sinensis* were <20% (Table 3). An important factor determining the behavior of seedlings is seed size (*Gross, 1984*; *Villar-Salvador et al., 2012*; *Larson et al., 2020*). The seeds of *L. clethroides* and *S. altissima* are small. Species with the largest seeds have the lowest mortality, especially in gaps, as large seeds enable quick initial growth, which is an important characteristic for establishment. Another explanation determining the behavior of seedlings would be the difference in initial allocation of biomass between above- and belowground parts. Fast development of the root system makes the seedling less vulnerable to topsoil desiccation (*Ryser, 1993*; *Leger, Atwater & James, 2019*; *Larson et al., 2020*). A small root system was probably one of the causes of the high mortality. *Solidago altissima* and *M. sinensis* produce more biomass aboveground than that in its root system, while *T. minus* var. *hypoleucum* produces more biomass for their root systems than that aboveground (*Yamada & Minami, 2015*; *Yamada, 2021*).

Monthly precipitation was unusually low in August 2020 (Table S1). The study site was bright conditions. Correspondingly, seedling mortality was greater on 9 September 2020

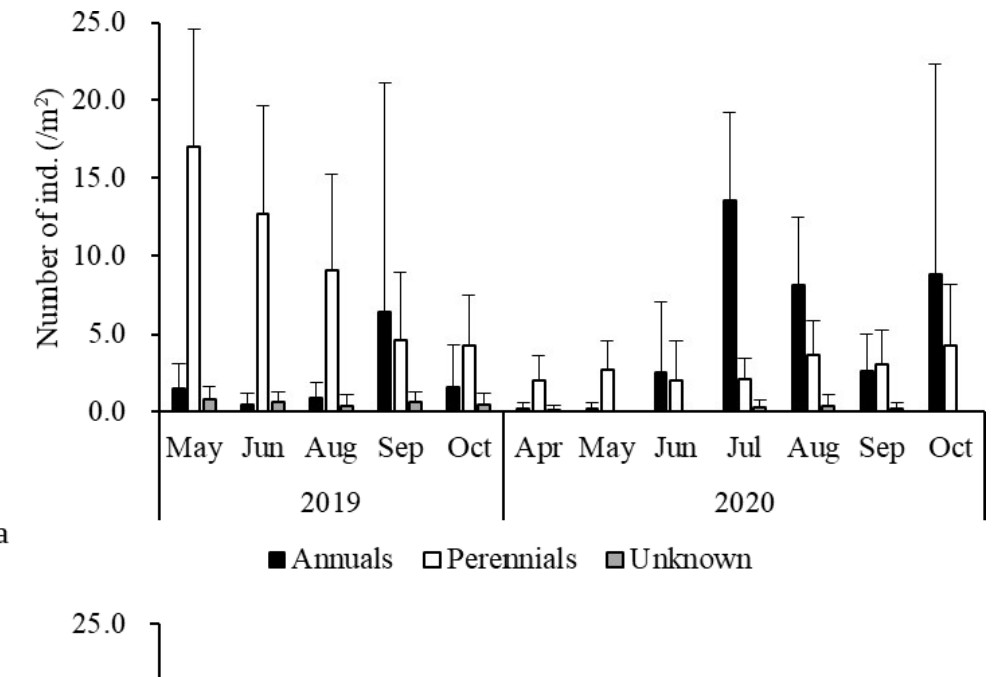

a

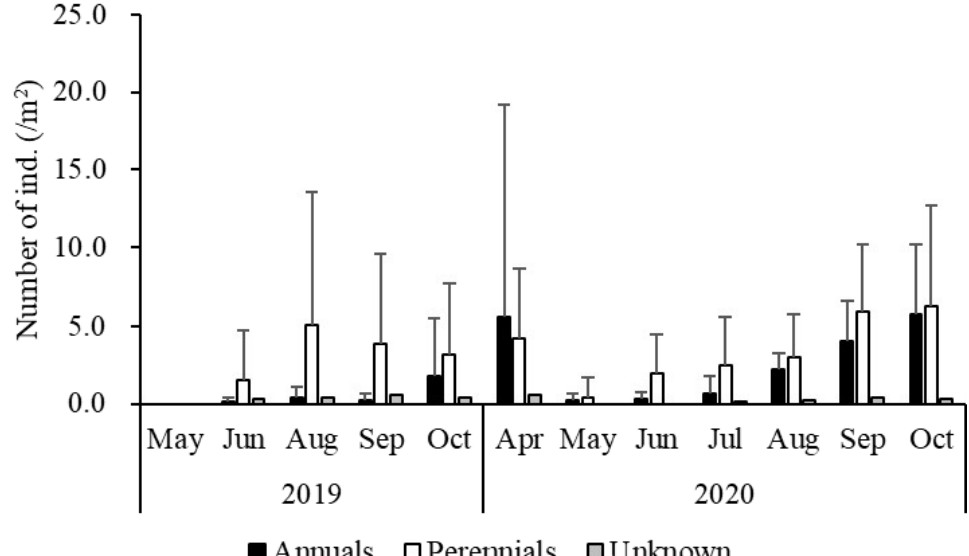

b

**Figure 5** (A) Emergence and (B) mortality of annuals and perennials in each period.

than on 5 August 2020 (Fig. 5). During summer, when the open gaps often dry up faster than the surrounding vegetation, seedlings may die owing to drought (*Martinkova, Honek & Pekar, 2009*; *Knappová, Knapp & Münzbergová, 2013*). Frost heave and drought in winter would also increase the mortality of the winter annuals *Veronica persica* and *Cerastium glomeratum* (*Ryser, 1993*; *Stampfli & Zeiter, 1999*).

No annual plants flowered (Table 4). Most annuals observed here were nitrophilous species (*i.e.,* favored by high nutrient content; *Török et al., 2008*) common in disturbed habitats. This grassland is characterized by a small number of annuals (Table 2; *Yamada*

*et al., 2005*), and reproductive failure would have contributed to this low number. Limited nutrient availability of the steep grassland likely led to the reproductive failure of the perennials, as it takes more than one year for young plants to reach the size enabling survival and reproduction in a harsh environment (*Špačková & Lepš, 2004*).

According to *Matsubayashi & Tamura (2005)*, the rate of soil creep in hilly area in Japan was up to several centimeters per year in slope of 30 degree. Considering severe steepness in the meadow, rate of soil creep may be severer and surface soil would be more unstable. Due to the unstableness, individuals are more likely to receive unprecedented soil coverage from upslopes or disappearance (soil disturbance). This is a main reason we were unable to find about a third of the tags at the site. We were unsure of whether the plants were still alive nearby without tags (just buried) or had died. Further researches are needed for the influence of the soil disturbances for the restoration outcomes.

## Generality of the present study

This study was carried out in one patch of vegetation, due to the landscape conditions we could not establish formal replicates of the grassland patches. The grassland is organized into relatively small, discrete linear elements because of the abrupt slope angle, which also makes the establishment of an ideal experimental layout impossible. Nevertheless, the vegetation in the study was similar to those reported elsewhere both in pre-abandonment (*Kitazawa & Ohsawa, 2002*) and during abandonment (*Yamada, 2011*) on such steep slope grasslands. Species disappeared in abandonment phase and re-appeared thereafter including species recorded in both shorter and longer abandoned plots without clear decline reported by *Yamada (2008)*. These features will decrease the specificity and increase the generality of the present study in this kind of landscape.

## Improving restoration of grasslands on steep slopes

Most typical grassland species in the Susogari grassland did not reach the reproductive stage by the second year of management reintroduction, and it remains unclear whether seedlings of such species will successfully reproduce in the long run. If individuals are growing, however, their tolerance of harsh environmental factors (*i.e.,* drought and frost) can become stronger over time. Nevertheless, vegetation cover was greater in 2020 than in 2019 (Table S3). Because of tall vegetation during abandonment, vegetation from upslopes would shade individuals on downslopes. This would prevent the rapid recovery of dominant species in 2019, the first year of management reintroduction. The increased number of seedlings and steady growth of vegetative propagules increased the cover in the second year. Although interspecific competition is less likely owing to the site's harsh conditions, the frequency and timing of mowing should be determined carefully to prevent overgrowth by dominant species, since the growth periods of the dominant species overlapped with those of other target species.

Many sprouts (vegetative regrowth) were observed in 2019, the first year of management reintroduction. For instance, more plots had sprouts of *P. freyniana*, *Barnardia japonica*, and *Sanguisorba officinalis* than in 2018. Sprouts, if any, were observed for 11 out of 14 typical grassland species (see Table S1). These indicates the importance of sprouts for their

quick recovery. In particular, sprouts are important for dominant matrix forming species since average frequency of their seedlings was relatively small. Besides, survival rate of *Miscanthus sinensis* was low (Table 3). Although vegetative propagules have potentially shorter viability in soil than seeds, they can play an important role in restoration outcomes (*Touzard et al., 2002*; *Matus, Tóthmérész & Papp, 2003*). For instance, they can contribute to rapid seed production in restored sites. Further studies are needed of the role of propagules in successful restoration at recently abandoned sites.

Richness of typical grassland species reached the reference level (Table 2, Fig. 2). Thus, this study revealed the successful restoration of a grassland species on a steep slope (Susogari grassland). Nevertheless, vegetation composition shown in NMDS remained difference at least for several plots between 2008 and 2020. Indeed, the numbers and frequencies of small species and ferns remained small until 2020 (Table 2, Fig. 2), but these species were observed in other nearby habitats (*Kitazawa & Ohsawa, 2002*). If suitable management can help safe sites to persist, migration of such species would be likely in the longer term.

## CONCLUSIONS

This study revealed the successful recovery of typical grassland species on a steep slope (Susogari grassland). Propagules (either seed bank or sprouts) of most species apparently did not diminish despite the high risk of soil runoff on the steep slope. Although several annuals and the invasive exotic perennial *S. altissima* appeared as seedlings, only a few were successful in producing seeds. By contrast, typical grassland species had relatively high survival rates in general. Thus, poor soil water and nutrient availability worked as strong filters to inhibit the establishment of annuals and exotic species, contributing to the formation of the characteristic species composition in the grassland. Severe soil disturbance (surface soil creep) should affect mortality of the seedlings, indeed, we lost track of about a third of the seedlings and their tags in our survey, but species occurring in the original grassland vegetation benefitted from the steepness of the slope overall, at least in the present site. It takes longer to restore degraded ecosystems (*Waldén & Lindborg, 2016*). Vegetation composition was not fully restored in the present study. The numbers and frequencies of small species and ferns remained small until 2020. We can expect further recovery of vegetation by continuous appropriate management which enhances the migration of absent species from nearby habitats.

## ACKNOWLEDGEMENTS

Two reviewers for helpful comments on the manuscript.

### Funding

This research was supported by JSPS KAKENHI Grant Number JP #20H03015. There was no additional external funding received for this study. The funders had no role in study design, data collection and analysis, decision to publish, or preparation of the manuscript.

## Grant Disclosures

The following grant information was disclosed by the authors:
JSPS KAKENHI: JP #20H03015.

## Competing Interests

The authors declare there are no competing interests.

## Author Contributions

- Susumu Yamada conceived and designed the experiments, performed the experiments, analyzed the data, prepared figures and/or tables, authored or reviewed drafts of the article, and approved the final draft.
- Wakana Yoshida performed the experiments, prepared figures and/or tables, and approved the final draft.
- Minori Iida performed the experiments, prepared figures and/or tables, and approved the final draft.
- Yoshiko Kitagawa performed the experiments, analyzed the data, authored or reviewed drafts of the article, and approved the final draft.
- Jonathan Mitchley analyzed the data, authored or reviewed drafts of the article, and approved the final draft.

## Field Study Permissions

The following information was supplied relating to field study approvals (i.e., approving body and any reference numbers):

Field experiments were approved by Tokyo Metropolitan Government, Japan.

## Data Availability

The raw measurements are available in the Supplementary Files.

## Supplemental Information

Supplemental information for this article can be found online at http://dx.doi.org/10.7717/peerj.17487#supplemental-information.

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
