# Peer review of "Fast grassland recovery from viable propagules after reintroducing traditional mowing management on a steep slope"

_PeerJ, doi:10.7717/peerj.17487_

## Round 0.1 · original submission · Major Revisions

Restoration ecology is an academic study that provides information on how to reverse the negative effects of human activities in nature. Every research endeavor aimed at unraveling the intricacies of conservation biology holds immense value. I am sure that the findings of your study will serve as a crucial resource for future researchers. However, it is essential to address some technical aspects in refining your article. I recommend a thorough review of the reviewers' suggestions and a judicious consideration of each recommendation. If you find yourself in disagreement with any particular suggestion, it would be beneficial to provide clear and well-reasoned justifications for your perspective.

·

Basic reporting

Abstract & Introduction

The introduction reads pretty well but I found myself a little confused by the history of these grasslands in the first three paragraphs. In particular, I was confused about how these grassland fragments were historically managed and their relationship to fully natural grassland. I don’t think this is a huge rewrite, the authors can improve these first few paragraphs by adding some clarifying points and some a few more sentences about how these systems were managed historically. Below I highlight some key questions I had to help show where I was confused.

I also have some recommendations on how to strengthen your hypothesis and better connect your research to the restoration field.

Line 21 – 25: Can you clarify if these semi-natural grasslands were traditionally maintained through mowing? This is never explicitly stated and the transition between the first and second sentences is a bit abrupt. You mention it in the Methods (lines 127-134). I would bring this into the introduction.
Line 42 -44: Can you define what you mean by semi-natural grassland? It appears like the sentence starting on line 43 is the definition but it is never explicitly stated.
Line 47 & 68: Can you clarify whether the species diversity in these grassland patches is associated with specialists or generalists? The statements you make in Line 47 and line 68 appear to contradict each other.
Lines 48- 51: Can you add a discussion about why you are targeting semi-natural grasslands as opposed to targeting/restoring fully natural grasslands. I just don’t understand why you would aim to restore the historic native community.
Lines 52-54 & 64-66: Can you better describe how these grassland fragments were managed within agricultural systems? You initially state that these grassland fragments were sub-optimal but later on state they were mowed. I just don’t understand why these fragments would have been maintained at all if they couldn’t be farmed. Relatedly, were these systems historically grazed before people?
Line 110 – 112: The way the current hypothesis is written is confusing. I would recommend rewording to something like “We hypothesized that reintroduction of mowing shortly after grassland abandonment would stimulate spontaneous recruitment of native species from the seed bank.
Line 112 – 115: I would focus these sentences less on what you would measure and instead describe what you would expect to find if your hypothesis is correct. This would be an excellent place to describe what you would consider success. Are you trying to get back to 2008 conditions? Before that? Right now it is unclear what a successful restoration project would look like.
Some additional minor questions/comments I had throughout the rest of the introduction.
Lines 76-80: I would add that spontaneous restoration still requires the natural disturbance regime to be reinitiated.
Lines 91-93: Is it important for conserving grassland species diversity in general or only within agricultural landscapes?
Lines 98-99: Recommend rewording slightly for grammar. “If the duration of abandonment is shorter, however, it is more likely that propagules will be viable in the seed bank.”
Lines 99-100: Wouldn’t the bigger point for restoration be, that if we can starting managing these systems while there is a viable seed bank restoration efforts will be less costly AND more likely to lead to successful restoration of native communities?
Lines 100 – 105: How does land abandonment interact this this? I would think that seed retention would increase after land abandonment due to the buildup of thatch…

Experimental design

Overall, I was able to follow the methods and understand what was done. But I have big concerns with the statistical approach of this paper and its lack of replication. Right now, based on what I can gather from Figure 1 and the description of the study design in the materials and methods this study appears to have an N=1 and the replicates the report below are actually pseudo-replicates.

I am sympathetic to the difficulty in getting multiple replicates for a restoration study. But if you only have one site you are going to be limited in what you can analyze statistically. If this is the case, I would recommend reframing this study as a case study with the 2008 data set essentially acting as your target or baseline condition. You can then talk about 1) how degraded the community was after not managing it for several years and 2) how close the community came back to 2008 conditions after you reinstated mowing in 2019. You could compare the communities at these discrete time points by looking at species richness, relative proportion of native/invasive, and how different the community is from the 2008 condition. I still think there is an interesting study here, but it is more about using this restoration project as a case study for other projects. Not a scientific experiment.

Some additional suggestions below

Figure 1: Could be more informative. I would add more detail to the map – for someone not familiar with Japan it isn’t very informative. I would add a panel of the map that is focused on more of the regional details.
Lines 127-134: I would add mention to Figure 1 photos. This seems like a good reference point for those photos.
Lines 122- 125: Was the annual precipitation any different or just the months noted? I think presenting the temperature and precipitation for the years of the study during the active growing months would be more useful than averages. This could be presented in a table for ease of reading.

Lines 156-158: Based on Figure 1 and this description it sounds like there is only one site. This would mean you have an n=1 and the quadrats you have laid out are actually pseudo-replication. Unfortunately, this makes it difficult to do any meaningful statistics because of the lack of replication. See above.

Line 160: Are primary leaves the Cotyledons? If so, I would use that term instead since it has a very specific meaning and it is well known that it can be difficult to identify a plant based on Cotyledons.

Lines 174-175: What caused the soil disturbance? I would like more information on this please. Was it a natural process?

Lines 190 – 213: After reading the entire Analysis section I would appreciate clarification on when you are using functional categories as opposed to species richness. You mention both but it is not clear if the NMDS or GLMs were conducted on species level data or on these broad functional groups. Or both? Broadly speaking I would recommend describing what you hope to answer with each of the analyses. What question you are hoping to answer that will help you evaluate your hypothesis.

Lines 219 -223: I would appreciate connecting the analysis section back to the hypothesis here at the end of the section. I would do this instead of generically talking about restoration projects. Alternatively, you could integrate the Ruiz-Jaen & Aide (2016) article into your hypothesis in the beginning so it doesn’t seem to come out to the blue down here.

Analysis Section – how did you use the soil data and the light data? Why did you collect that information?

Validity of the findings

Results

I have some broad overarching comments and suggestions for improving this section. But my concerns about the statistical validity of the methods, described above need to be addressed before I can fully review this section.

I recommend restructuring your results section so that it better tells the story you are trying to tell. I would recommend first describing the original baseline conditions in 2008, then in 2018 after abandonment, and then finally presenting the data from after the mowing was restarted. By restructuring the results section this way, it would be easy for the reader to understand how the vegetation community had changed and how effective mowing was at restoring the initial condition.

I would also add data about the invasive vs native species and how their proportions shifted over time. This is such a central challenge in restoration and for this type of restoration activity that I think it would greatly add to the conversation. Relatedly, I would like to see results about the diversity of the communities first before diving into the specific species/functional groups.

I am confused about why you have separated established vegetation vs seedlings. To me these are two different components of the plant community and need to be discussed together. If you want to keep them separate, you should better explain in the methods about gathering data on the established vegetation vs seedlings. Also you should explain why it is important to consider these two aspects of the community separately from one another in the introduction.

Lines 230 & 237 (Figure 2 & 3 as well): I would refer to the different classes of plants as “functional groups” instead of attribute or behavioral types. This is a standard way to refer to this was to classify plants.


I also have a few thoughts and suggestions on ways to improve, but it was not possible to do a thorough review of this section due to the concerns I highlighted above.

Lines 268-277: I would move this paragraph to later in the discussion. I think it is always better to start with the big takeaway before highlighting the limitations of your study

Reviewer 2 ·

Basic reporting

Important research in a rare grassland ecosystem with clear management implications. Understandable and clear writing with some minor word choices that could be clarified. Literature cited is a decade old and could be updated with more recent papers. Figures and tables provide support for their arguments but the caption and axes could be clarified.

Experimental design

The study design is simple and appropriate for the questions they are asking. The only concern is that the sampling was only done on ten quadrats, which is ok for this size of study area but not enough to describe the whole Susogari grassland.

Validity of the findings

The statistical analyses are sound and clearly stated. The novelty of this study is a long-term assessment of seedling recruitment after mowing in this ecosystem. Results could be better described linking to pre and post treatment.

Additional comments

Major:
1. Citations could be updated throughout the manuscript as they are mostly a decade old. There are more recent work on passive restoration, demography, and native plant establishment that could be added in the intro.
2. The results on established vegetation is difficult to interpret as written. If this section is rewritten as change in dominant species and functional groups pre- and post- reintroduction of mowing in 2018, I think it would be easier to understand.
3. Throughout the results and discussion, which species are common and rare could be reminded with parentheses. For a reader unfamiliar with the species in this grassland ecosystem, it is not clear which species are desirable for mangers here. Do they want the common grassland species? Are there any species of conservation concern?

Minor:
Line 42: I am not familiar with the term semi-natural grasslands. Could you define it here?
Line 45-48: I am confused by this sentence. Is habitat generalist and non-specialist the same? Why does colonization or input of generalist species result in unique composition?
Line 77: I would consider rewording “spontaneous restoration” to “natural recovery” or “spontaneous succession” to be consistent with Prach 2003.
Line 81: I don’t think there is much ambiguity about the vulnerability of seedlings. I would be more direct here and remove “probably,” and “In terms of the availability…” clause isn’t needed.
Line 83-85: More recent papers could be cited here. For example, see Larson et al. (2020) Ecological strategies begin at germination: Traits, plasticity and survival in the first 4 days of plant life. Functional Ecology. 34(5): 968-979.
Line 86: High mortality from what? Frost? Drying?
Line 89: Consider changing ‘recreated’ to ‘re-established.’
Line 110: Do you mean seeds in the seedbank by ‘most propagules’?
Line 112: What is seed runoff? Seed rain from surrounding areas? This is making me think that colonization from surrounding areas is bad. Is it bad because those surrounding areas have non-native species in them?
Line 158: How was this transect oriented? Was it a 10 m transect?
Line 237: Change “behavioral types” to “functional groups.” I would be interested in reading about which functional groups declined or increased from treatment.
Line 283: Remind the readers which species are present in typical grasslands here? Carex and Cirsium?
Line 323: Change “didn’t” to “did not.”
Line 357: Change 1 to one.
Line 363: What do you mean by unprecedented soil coverage? Do you mean significant soil erosion?
Line 369: Which species are they? Maybe mention “Most typical grassland species in Susogari grassland…” to remind the readers which ecosystem you are talking about.
Table 1: Why is there no dominant species in 2019?
Figure 2: a) total of what? Species richness? What is the unit on the y axis? Is the y axis species richness or change in species richness? If it’s a change in species richness, change from what and why isn’t there a negative change?
Figure 3: Same as Fig 2, change from what? Add error bars.

---

## Round 0.2 · accepted · Accept

I would like to thank you for accepting the referees' suggestions and improving your article based on their input. I believe your manuscript is now ready for publication. We look forward to your next article.